# Emerging Risks from Embodied AI Require Urgent Policy Action

**Jared Perlo**
Centre for the Governance of AI
French Center for AI Safety (CeSIA)

**Alexander Robey**
Carnegie Mellon University

**Fazl Barez**
University of Oxford
WhiteBox

**Jakob Mökander**
Tony Blair Institute for Global Change
Yale Digital Ethics Center

## Abstract

The field of embodied AI (EAI) is rapidly advancing. Unlike virtual AI, EAI systems can exist in, learn from, reason about, and act in the physical world. With recent advances in AI and hardware research and design, EAI systems are becoming increasingly capable across an expanding set of operational domains. While EAI systems can offer many benefits, they also pose significant short- and long-term risks, including physical harm, surveillance, and societal disruption. These risks require urgent attention from policymakers, as existing policies for industrial robots and autonomous vehicles are insufficient to manage the full range of concerns EAI systems present. To address this issue, this paper makes three contributions. First, we provide a taxonomy of the physical, informational, economic, and social risks EAI systems pose. Second, we analyze policies in the US, UK, and EU to assess how existing frameworks address these risks and to identify critical gaps. We conclude by offering policy recommendations for the safe and beneficial deployment of EAI systems, such as mandatory testing and certification schemes, clarified liability frameworks, and strategies to manage EAI's potentially transformative economic and societal impacts.

## 1 Introduction

Embodied AI (EAI) refers to artificial intelligence (AI) systems and agents that are grounded in the physical world and learn through perception and action [1, 2]. EAI systems can operate across diverse environments. For example, existing EAI applications can deliver packages [3], patrol public spaces as security guards [4], or care for humans in elder-care homes [5, 6]. EAI capabilities and domains are likely to expand significantly in the coming years [7, 8]—presenting both opportunities and risks for humans. While EAI systems already assist people with mobility impairments in navigating the world (e.g., autonomous cars), future systems could fill agricultural or manufacturing jobs as working-age populations decline. By augmenting and complementing human labor, EAI could foster economic development and prosperity [9]. On the other hand, EAI systems can inflict more immediate physical damage than virtual AI systems and may also cause social harm as humans form closer connections with these systems [10, 11]. See Figure 1 for a comparison of classical robots, agentic AI, and EAI.

Recent breakthroughs in AI capabilities—particularly those related to Large Language Models (LLMs) and Large Multimodal Models (LMMs)—have catalyzed unprecedented progress in EAI systems' ability to navigate and act in the physical world [12, 13]. At the same time, the rise of Vision-Language-Action Models (VLAs)—which cast control as next-token prediction over visual and linguistic tokens—opens the possibility for a "ChatGPT moment" for robotics, with sharp jumps

39th Conference on Neural Information Processing Systems (NeurIPS 2025) Position Paper Track.

in capability and public awareness. Recent debuts of models like Gemini Robotics-ER and NVIDIA's Isaac GR00T N1 marked significant EAI algorithmic progress, even though these models are only slowly being paired with hardware advanced enough to translate virtual capabilities into real-world actions [13–15]. In the past few months, for example, EAI systems have completed half-marathons and shown the ability to unpack groceries with little prior context [16, 17], and open-source resources from companies like Physical Intelligence and Unitree could spur continued progress [18, 19].

Data acquisition—a bottleneck for EAI development due to the complexity of information needed to train models [20]—is being addressed through open-source datasets and cross-modality approaches [21]. Simultaneously, innovations in tactile sensing, LiDAR, actuators, and power systems are expanding the potential capabilities of EAI systems [22–24]. Progress in physical abilities and data collection lower barriers to creating high-quality models about how the world operates [25]. These world models involve perception, reasoning, and memory [26], and increasing EAI research funding could lead to more accurate world models and positive development feedback loops. EAI research is also emerging as a new frontier in geopolitics, as concerns about supply chains and national industrial policy become more salient [27, 28].

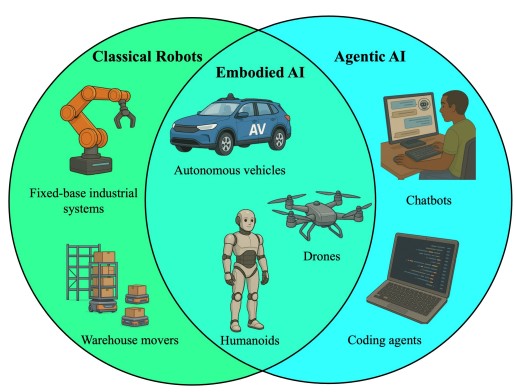

Figure 1: **Comparing classical robots, agentic AI, and EAI.** EAI combines the autonomy and reasoning capabilities of agentic AI with classical robots' physical embodiment.

EAI's advancing capabilities are giving rise to many short- and long-term risks. Although EAI shares many traits with virtual agents [29], such as varying degrees of autonomy and capability [30], the physical embodiment of EAI systems introduces distinct considerations that warrant special attention [31]. EAI systems can hit, cut, bump, attack, and more, whether intentionally or not. The physical world presents significant adaptation challenges for models trained in virtual simulations [32, 33]. Technological breakthroughs may also increase scalability and later enable self-improvement, allowing EAI to rapidly advance and face fewer human bottlenecks [34]. Social, legal, and economic systems will likely require significant updates [8, 35].

Before rushing to policy action, it should be noted that EAI is not new but an evolution of traditional robotics. The EAI field builds upon decades of science fiction imagination, human-robot interaction research, and forecasting about advanced robotics [36–38]. In fact, the term "embodied AI" is partly a marketing technique used to differentiate recent innovations from traditional robotics.

Safety concerns about EAI are likewise not novel, as researchers have studied safety in robotics for decades [39, 40]. Tools to formally verify robot behavior have included model predictive control [41], control barrier functions [42], and temporal logic [43]. Many papers focusing on safe AI design prominently feature imaginary robots as examples of human-AI collaboration [44, 45]. More recent work has focused on creating safety guardrails for robots from real-world data, such as in Sermanet et al. [46]. Still, beyond a UN resolution initiating lethal autonomous weapon discussions [47], there largely remains a policy vacuum regarding EAI safety at national and international levels.

Understanding and minimizing risks from EAI will become even more critical in a world with AI capabilities equivalent to or surpassing artificial general intelligence (AGI), however defined [48]. AGI uncertainties aside, EAI risks are critically understudied and poorly understood, and current regulatory frameworks are generally insufficient to guide safe EAI development. This paper clarifies the risks and governance challenges posed by EAI and suggests a pragmatic sociotechnical approach to help governments and researchers support the development of safe EAI [49]. This paper makes three unique contributions to address this urgent issue:

1. We develop a comprehensive taxonomy of risks from EAI, spanning physical, informational, economic, and social dimensions. This taxonomy covers concerns ranging from malicious physical harm from jailbreaking LLMs and privacy violations in homes to widespread labor displacement. To create this taxonomy, we draw on the literature related to robot safety, human-robot interaction, and recent predictions about AI's trajectory.

2. We analyze existing EAI policy frameworks to assess their adequacy and highlight critical gaps. Although specific pieces of legislation governing autonomous vehicles or advanced robotics trend in the right direction, significant gaps remain. For example, current robotic regulations are ill-suited to govern systems that have high levels of autonomy and continuous learning; these characteristics challenge existing safety testing and assurance paradigms.
3. With these risks in mind, we discuss several policy interventions to improve EAI safety. We suggest increasing targeted safety research, establishing robust certification requirements for EAI, promoting industry standards, clarifying liability regimes, and creating actionable policy blueprints to respond to transformative economic and social effects of EAI.

This paper has several limitations. We focus on civilian applications of EAI, although military applications also merit consideration. We focus on frameworks from the US, UK, and EU, though emerging regulatory efforts in other regions deserve increased attention. Ensuring safe EAI will also require a multi-layered approach, with mechanisms to enhance safety at the model, application, and organizational levels [50]. While it remains crucial to ensure the safety of underlying models through AI safety research, we focus on strengthening safety measures for EAI-specific applications. Acknowledging this context, we aim to provide a solid foundation upon which future work can build.

In the coming years, policymakers may quickly become aware of the risks posed by EAI because of headline-grabbing breakthroughs. This could rapidly elevate EAI regulation on policy agendas, so policymakers must be equipped with appropriate context to create clear and beneficial legislation. To ensure the safe and beneficial development of EAI, we argue that policymakers must urgently build upon and address gaps in existing frameworks for robotics, autonomous vehicles, and agentic AI.

## 2 Taxonomy of Risks from EAI

Drawing on existing research and predictions about EAI trajectories, we identify four crucial areas of EAI risks: physical, informational, economic, and social (see Figure 2). This taxonomy leads to our discussion of how existing policy frameworks address—or fail to address—these EAI risks.

### 2.1 Physical risks

**Purposeful or malicious harm.** EAI systems present several distinct physical risks. EAI systems have been designed and deployed with lethal intent, such as AI-controlled drones [51, 52]. However, fully autonomous military robots, often integrated with bespoke AI architectures [53–55], are not yet widely used in combat. Commercially available EAI systems, including AI-controlled quadrupeds and autonomous driving assistants, also present serious risk. Recent research has demonstrated that these systems inherit *jailbreaking* vulnerabilities from LLM-based AI models [56–59]. This could allow malicious actors to subvert safety guardrails and perform a range of harmful physical tasks, including detonating explosives and causing collisions [60–62]. VLAs exacerbate this risk: an attacker might craft a visual scene or textual instruction that, when interpreted through a language-action policy, yields physically dangerous instructions not anticipated by vision- or language-only defenses [63, 64].

**Accidental harm.** Automation in sectors ranging from manufacturing to healthcare will put humans into close contact with EAI systems [7]. This interaction increases the risk of accidental physical harm. Several recent reports document an increase in industrial injuries following the introduction of AI-controlled robots [65–67]. EAI systems could accidentally cause physical harm through misspecified goals, physical hardware malfunctions, or other unanticipated behaviors [44, 68, 69]. For example, a humanoid EAI might not correctly reason that placing a full glass of milk on a tilted table is perilous and likely to lead to a dangerous broken glass [70]. Researchers also face persistent difficulty in getting models trained in purely virtual simulations to act as intended in the real physical world—what is referred to as the "reality gap" [71]. This introduces significant scope for accidental harm if the deployed world does not closely match an EAI's training data [72].

### 2.2 Informational risks

**Privacy violations.** EAI systems are often trained on vast corpora and process a variety of data modalities—spanning visual, auditory, and tactile information—during deployment, creating significant privacy concerns [12]. Like text-based virtual AI models, which are known to memorize and expose personally identifiable information [73, 74], commercial robots have been shown to disclose

| Physical | Informational | Economic | Social |
|---|---|---|---|
| | | | |
| **Key Risks** | Purposeful harm, accidental harm | Privacy concerns, misinformation | Labor displacement, socioeconomic inequality, power concentration | Lack of trust & transparency, unclear liability standards, bias & discrimination |
| **Example Mechanisms** | Jailbreaking, sensor spoofing, hardware malfunctions | Unauthorized surveillance, non-consensual data usage, hallucinations | Recursive EAI design, rent-seeking behavior, uneven model diffusion | Human-robot romance, EAI therapists, doctors, etc., hiring bias toward EAIs |

Figure 2: **A summary of risks from EAI**. We identify four key risk categories and provide several existing or potential mechanisms by which EAI systems could cause harm within each category.

proprietary information through simple prompts [60]. EAI's mobility and the vast array of sensors used in EAI technologies expand concerns about unauthorized data collection. For example, EAI systems can monitor user behavior, infer physical preferences, and contribute to future model training without the consent of those being observed beyond the limitations of immobile microphones or security cameras [75–77]. Bad actors could gain access to private data streams and monitor users' movements, providing leverage over individuals to squash dissent or seek power [78].

**Misinformation.** Several studies have shown that LLMs often hallucinate information [79–81]. EAI systems inherit these shortcomings in the physical world, answering user questions with deceptive or incorrect information [82]. Because VLAs fuse vision and language, their hallucinations can be spatially grounded—e.g., misidentifying an object in view and then generating a plausible yet unsafe plan around it. And although automated assistants like Amazon's Alexa are known to lie about issues as innocuous as Santa Claus [83], more capable and trusted EAI systems in sensitive positions (like home-assistant or community-service positions) could easily spread model developers' propaganda to users. For example, an EAI running on DeepSeek's latest model could provide a subtle yet continuous stream of misinformation to American users [84, 85].

## 2.3 Economic risks

**Labor displacement.** While virtual AI applications will likely displace certain types of human cognitive labor, EAI systems could significantly replace or displace physical human labor [86]. At a minimum, EAI will likely augment the type of work that humans perform [87, 88]. Classical industrial robots have taken over many human roles in manufacturing [89], and research has shown that robot deployment can reduce human employment [90]. Future technological advances will likely accelerate this displacement, as increasingly capable EAI systems perform complex physical tasks beyond assembly lines. Though automation has historically redirected labor toward areas of human comparative advantage [91], AGI-enabled EAI could potentially automate all physical labor [92].

**Socioeconomic inequality.** EAI could significantly exacerbate wealth inequalities. Those who have access to or own EAI systems will be able to automate labor and perform tasks significantly better or faster than those without access. These significant productivity advantages will potentially concentrate wealth and exacerbate domestic and international inequality [93, 94]. For example, while a wealthy businesswoman could invest in a fleet of the latest humanoid robots, individuals lacking adequate capital might be forced to rent their EAI systems [95]. Virtual AI applications may cause similar socioeconomic inequality, but the ability to control access to EAI systems may confer unique returns on investment, given that many physical tasks necessary for human survival (e.g., growing food, building shelter) are constrained by human strength and energy.

**Power concentration.** EAI deployment could accelerate the consolidation of economic and political power. By unlocking increasing returns to capital, EAI will decrease employers' reliance on the needs of human labor [96]. EAI users or consumers may become dependent on EAI owners for goods and services due to relevant productivity advantages [97, 98]. The importance of EAI to perform physical tasks will likely exacerbate power-concentration risks presented by purely virtual AI systems. The proliferation of EAI systems could thus lead to a rapid concentration of corporate economic (and social) power, potentially even facilitating an eventual coup involving EAI [78, 99].

## 2.4 Social risks

**Bias and discrimination.** Like virtual applications of AI, EAI can discriminate against users. When EAI systems are placed in positions of power, their biases could have significant impacts on fairness in everyday interactions [100, 101]. For example, a peacekeeping humanoid robot may discriminate based on skin color [102]. Unlike virtual AI applications, this bias can have immediate and irreversible physical consequences (e.g., if the peacekeeping robot mistakenly injures an innocent passerby).

**Lack of accountability and liability.** Determining liability when EAI causes harm requires new frameworks that address the complexities of autonomous physical systems. Human users may disagree with decisions taken by EAI systems, raising questions of delegation and responsibility [103]. Lack of EAI accountability could lead to confusion for users and breakdowns in traditional justice systems [104]. For example, we may soon need to consider who to blame and how to collect damages when a highly autonomous robotic surgeon removes a healthy organ by mistake [105].

**Lack of transparency and explainability.** Understanding how AI systems reason and why they perform specific actions motivates the field of interpretability research [106]. But physical embodiment raises the stakes for understanding these systems. For example, transparency of planned actions and explainability of decision-making is crucial when an AV suddenly changes lanes. A lack of transparency could lead to a lack of trust, which could become a socially destabilizing issue with widespread EAI deployment [107–109].

**Unhealthy human-EAI relationships.** Interactions with EAI systems could foster dangerous human dependence or romantic attachment [110]. People may depend on EAI systems for physical pleasure [111]. The presence and human-like features of EAI systems may amplify the dependency issues already observed with conversational AI [112, 113]. People may easily fall in love with EAI systems, only to be distraught when these systems are altered or have their memories reset [114].

**Transformative effects.** EAI deployment could reshape society, particularly if technological development outpaces society's ability to adapt [98, 115]. For example, EAI systems could provide physical threats of violence or mass surveillance to back up AI-enabled authoritarianism [116]. Businesses might prefer to employ EAI systems, affecting how humans find meaning in their work [117]. Humans might also lose the ability to perform various jobs as tasks are delegated to EAI systems [96].

## 3 Heat map of relevant policies

Existing policy frameworks address many risks identified in §2. Understanding how current regulations apply—or fail to apply—to EAI systems is essential for both policymakers and researchers. This section examines key policies from the US, UK, and EU that govern related technologies, including classical robotics, autonomous vehicles, and virtual agentic AI. Our analysis identifies regulatory gaps specific to EAI by examining where existing frameworks provide minimal, adequate, or substantial policy coverage. This non-exhaustive review focuses on civilian applications of EAI.

### 3.1 Key policies

This section first maps policy frameworks that govern physical harms, focusing on laws and standards that apply to AVs and robots, before examining major pieces of legislation that address informational, economic, and social EAI harms.

**Physical risks.** Existing governance approaches to physical risks posed by EAI mainly target AVs and drones. AV-specific laws follow one of two pathways: creating bespoke legislation or adapting conventional automobile laws [118, 119]. For example, the UK's Automated Vehicles Act 2024 introduced the concept of the Authorized Self-Driving Entity (ASDE) and the No-User-in-Charge

Table 1: **Coverage of policies for major EAI risks.** We examine whether existing governance frameworks address risks from technologies related to EAI. ● indicates that there is a high level of coverage of relevant policies; ◑ indicates there is partial coverage but that significant adjustments are necessary; ○ indicates a significant lack of governance frameworks to address the relevant risk. We reference AVs rather than broader EAI, as most EAI regulations to date have addressed AVs.

| Risk | Subrisk | Classic robots | AVs | Virtual agents |
|---|---|---|---|---|
| Physical | Purposeful or malicious harm | ● | ◑ | ◑ |
| | Accidental harm | ● | ● | ◑ |
| Informational | Privacy violations | ◑ | ◑ | ◑ |
| | Misinformation | ◑ | ○ | ◑ |
| Economic | Labor displacement | ○ | ○ | ○ |
| | Socioeconomic inequality | ○ | ○ | ○ |
| | Power concentration | ○ | ○ | ○ |
| Social | Bias and discrimination | ◑ | ◑ | ◑ |
| | Accountability and liability | ◑ | ◑ | ◑ |
| | Trust and transparency | ◑ | ◑ | ◑ |
| | Human-EAI attachment | ○ | ○ | ◑ |
| | Transformative societal effects | ○ | ○ | ○ |

(NUiC) operator. These entities (e.g., manufacturers or fleet operators) assume legal liability when the vehicle is in self-driving mode, effectively standing in for the "driver" [120]. These ASDE and NUiC entities could serve as precedents for other forms of highly autonomous EAI; however, these roles largely fit neatly into existing automotive regulatory structures. Different forms of EAI—such as home care or educational EAI—will not have the same pre-existing foundation.

In the US, the recently proposed ADS-equipped Vehicle Safety, Transparency, and Evaluation Program (AV STEP) would require approved AV manufacturers to share details about vehicles' development and operation. This would include information about simulations used to train AVs' algorithms, environments in which the AVs are designed to operate, and oversight mechanisms to ensure operational safety [121]. AV STEP is a promising framework that could be extended to other EAI contexts, though it is unclear which regulatory body would oversee other EAI modalities [122].

Laws concerning aerial drones are also relevant to EAI, although many of today's drones operate with limited autonomy [123]. EU regulation on drones distinguishes between remotely-piloted drones, fully autonomous drones, and drones that fly pre-planned routes [124]. The EU requires that all drones are operated by a remote pilot who assumes responsibility for each flight. Pilots of autonomous and other high-risk drones must pass knowledge and practical-skill courses. However, National Aviation Authorities often depend on pilots' self-reporting to verify that drone operations pose minimal risk, for example by confirming pilots avoid flying over people "uninvolved" in the drone's operation. Non-autonomous drones require less stringent training to operate but face operating restrictions; for example, pilots of these drones must always keep the aircraft within their line of sight.

Another key piece of legislation is the EU's Machinery Regulation (MR), passed in 2023 [125]. Updating similar legislation from 2006, the MR regulates many types of robots in the EU and explicitly addresses aspects of AI and EAI safety, as discussed by Tobias Mahler [126]. The MR, which addresses issues ranging from emergency-stopping systems to the risk of being trapped inside a device, mentions machines with "self-evolving behaviour." The MR mandates that machines sold in the EU must be tested for compliance with these safety regulations; as with other EU regulations, third-party evaluators (or "notified bodies") test whether machines fulfill the safety requirements.

International standards also provide manufacturers with robust guidance for robotics. ISO 10218:2025 recommends safety protocols for assessing risk (e.g., through controls, safety and stopping functions) and safe-design certification for industrial robots [127]. In addition, ISO standards on service robots emphasize sensor reliability, uncertainty management protocols, and decision verification through multiple sensing modalities [128]. Many standards for AVs also already exist. For example, ISO/SAE 21434:2021 and UN Regulation 155 provide cybersecurity standards for road vehicles [129, 130]. These standards are particularly relevant for many forms of mobile EAI, especially those deployed in situations without regular access to trusted networks [131] (e.g., during disaster rescue missions).

**Informational risks.** Many informational risks apply to both virtual and embodied AI. Key frameworks governing EAI informational risks include the EU's AI Act and the General Data Protection

Regulation (GDPR). The AI Act prohibits several data practices relevant to EAI, such as untargeted facial image scraping and manipulative decision-influencing techniques [132].

GDPR legislation in the EU and UK establishes strict governance requirements concerning the capture, use, and storage of data. GDPR requires that data is only collected for "legitimate interests" and that entities collecting data are classified as "data controllers" who must document data collection practices while implementing robust security measures [133]. However, EAI deployments in public spaces challenge traditional consent models and controller identification [134]. EAI systems deployed in public spaces, for example, raise questions about who receives and controls data from system sensors, how to opt out of data collection, and even what constitutes a public vs. private space [135].

**Economic risks.** Regulatory frameworks to manage EAI's economic impact remain underdeveloped. Existing legislation, like the UK's Employment Rights Act 1996 and the US WARN Act, provides limited protections for workers facing technological displacement [136, 137]. Labor organizations have achieved isolated victories, exemplified by the International Longshoreman Association's successful challenge to port automation [138]. Policymakers must prepare for labor displacement resulting from EAI deployment in industrial settings, keeping in mind that EAI innovation could lead to a period of rapid creative economic destruction [139].

Some observers think AI development represents a different kind of technological transition compared to previous transformations, as AI may replace cognitive tasks in addition to physical labor [8, 140]. Economic policies may not need to target technological failures, as with many of the physical and informational risks mentioned in §2. Economic risks may instead emerge because EAI systems work *too* well, rapidly upending labor structures and reducing the need for human labor. As a result, policymakers should consider social policies to manage these emerging tensions [141].

**Social risks.** Few regulations directly address the social impacts of EAI. Those that do largely govern issues of direct human interaction with EAI and do not address larger issues of how society will transform as these systems become increasingly prevalent and powerful. The EU AI Act's broad prohibition on infringing fundamental rights could be extended to address EAI issues such as lack of trust, lack of transparency, unhealthy attachments, and bias and discrimination, but this would require further specification. In terms of accountability, proposed frameworks for attributing actions and delegating authority to virtual agents could prove helpful for EAI [142, 143].

GDPR Article 22 provides an instructive example of existing regulation that implicates but does not directly address EAI systems. Designed with virtual AI in mind, the Article prohibits individuals from being "subject to a decision based solely on automated processing" when that decision has legal consequences for that individual [133]. Yet it remains unclear how this Article could be reconciled with fully autonomous EAI, or how individuals could appeal to a human intervener—as the Article mandates—in immediate physical interactions or conflicts with EAI systems.

Beyond legislation, international standards for manufacturers and developers emphasize transparency, ethical design, and trustworthiness in EAI systems. But these standards, including the IEEE's 7000 series on autonomous system transparency [144], algorithmic bias [145], and the impact of robotics on human well-being [146], are voluntary. These standards could apply to EAI applications too, but their voluntary nature could similarly limit their impact.

## 3.2 What are the most significant gaps?

Though major building blocks to address harm from EAI systems already exist, several key policy gaps concerning EAI safety require urgent attention.

First, there are many open questions concerning robust certification for different EAI modalities. Regulating AVs is relatively straightforward due to defined operational domains (e.g., cars usually stay on roads). Future EAI systems, however, will likely have expanded freedom of movement, enabling them to enter residences and conduct surveillance in schools or public areas. This will require new processes to certify EAI systems' safety, and such frameworks do not yet exist. Expecting existing consumer-safety labs, which currently test the safety of machine components, to evaluate the safety of wider EAI systems is unrealistic. Basic questions such as identifying the relevant regulator are a key starting point—to what extent should there be EAI-specific consumer protection boards with AI expertise, or should existing third-party testing laboratories take on this responsibility?

Secondly, once a suitable apparatus is in place, EAI capabilities should be measured with reliable evaluations. To date, few of these benchmarks exist [46], despite the existence of many benchmarks for virtual AI systems [147]. Evaluations could cover a range of considerations outlined in our taxonomy, e.g., evaluating the robustness of the reality gap, defenses against jailbreaking, alignment between software and hardware capabilities, and hardware durability, among other areas.

Thirdly, policies currently devoted to post-deployment EAI monitoring lack detail. Such oversight mechanisms have been proposed for other AI systems [148]. However, current regulations requiring EAI systems to include "black boxes" that record and preserve data in the event of accidents or misuse are hazy. The EU's MR mandates data about safety-related decision-making processes is kept for a year after collection [125]. At the same time, the bloc's AI Act requires high-risk AI systems to retain this information for at least six months [132]. Such recording systems, in addition to live data monitoring, can enhance system safety and aid post-incident investigations [149, 150]. The EU's MR also states "it shall be possible at all times to correct the machinery...to maintain its inherent safety," but this oversight effort requires clarification for highly autonomous systems. Does this notion require the ability to tweak EAI actions in real-time, send model updates over-the-air, or another intervention? Are users, the government, or the manufacturer meant to perform this monitoring?

There are also gaps in policies addressing economic and social effects from EAI. Although EAI could cause labor displacement, proposals to distribute economic proceeds are still in their infancy [151, 152], as are national and regional proposals to ensure that world economies have sovereign systems (e.g., data centers, energy production, EAI hardware) to capture EAI's benefits. Policies addressing social issues related to trust and human-EAI attachment are likewise currently scant [153]. More broadly, there are significant policy gaps at the intersection of EAI and potential AGI. For example, should an EAI system be allowed to build other EAI systems? Should a country developing AGI in embodied form automatically and freely share the technology with the rest of the world? Policymakers must consider how EAI should be developed, deployed, and integrated into societal structures to address the broad array of neglected challenges mentioned here.

## 4 Proposed pathways forward

### 4.1 Invest in EAI safety research

We recommend increased research be devoted to EAI safety based on the risk taxonomy described in §2. For example, robotics and machine-learning researchers can further efforts to make hardware actuators less susceptible to hacking and malfunction through physical design and formal methods [154]. Building benchmarks and evaluations of EAI capabilities and behavior is a particularly promising area of EAI safety research. Most AI benchmarks today specifically target the virtual aspects of AI [155], although recent progress has been made in EAI research [46]. Researchers should and build on this progress by developing EAI evaluations and benchmarks that span a broad set of tasks and task types [156, 157], similar to the work being done by the RoboArena team [158]. Beyond physical risks, benchmarks and evaluations should also address issues related to privacy and cybersecurity, for example by building upon zero-knowledge proof research from other AI domains [159]. We also need benchmarks that stress-test the joint vision–language–action loop—measuring, for instance, whether a VLA model's visual prompt leads to safe, context-aware behavior across edge cases. Benchmarks and assessments will not address every risk raised above—particularly socioeconomic considerations—but they are a critical step towards minimizing many risks from EAI.

### 4.2 Create robust certification requirements before EAI deployment

National bodies should mandate that EAI systems pass safety evaluations and are certified for public use. EAI systems should have clear "model cards" describing how they were trained, in which domains they were designed to operate, and what safety measures the manufacturer has taken to ensure safe operation. Policymakers could then mandate that EAI systems be limited to legal operation within the specified domains, potentially aided by remote identification requirements similar to those for drones. This model card approach could borrow from the frontier safety frameworks that many leading AI labs have implemented [160]. This regime could be enforced via audits of EAI manufacturers and developers [161]. Policymakers should also ensure that this certification regime incorporates different categories of requirements based on potential risk [48]. For example, certifying EAI safety for children's toys or an autonomous limousine should involve different safety

testing requirements. EAI regulation should address concerns at the developer, model, and application layers, much like approaches for non-embodied AI [161]. Combining model- and application-specific approaches with policy efforts at the developer layer could help ensure robust and durable EAI safety.

## 4.3  Promote industry-led standards to address EAI risks

Standard bodies should push forward EAI safety efforts in tandem with legislative approaches. These standards bodies should both update existing standards and create new, dynamic standards for highly autonomous EAI. Existing standards are grounded in today's robotic capabilities, and even recent updates fail to address how EAI systems capable of advanced reasoning will affect real-world applications. In May 2025, the ISO announced promising intentions to create a new standard for humanoid robots, which should encompass additional form factors, notions of autonomy, and use cases [162]. Given EAI's fast-evolving nature, industrial actors can leverage their technical expertise to help develop and adjust standards more rapidly than international standards-setting bodies [163]. These standards should address technical protocols (e.g. for cybersecurity), mandate "black boxes" that record sensor input and reasoning preceding adverse events, and span the entire EAI continuum—from components to systems and swarms [164–166]. The tension between accountability and privacy raised by recording systems should be acknowledged and addressed in future standards.

## 4.4  Clarify liability regimes for fully autonomous systems

Policymakers should clarify existing, muddy liability regimes. When autonomous EAI systems are deployed in the near future, who should be held accountable for injuries or misuse? Should the person who gave the model its latest instruction be at fault, or should the blame rest with the manufacturer? In the longer-term, how should EAI systems themselves be held accountable for faults if they are considered fully automated and agentic? EAI liability is a growing area of legal study [37, 167–169], but firm policies need to replace current ad-hoc legal approaches. For instance, policymakers should clearly define notions like the Authorised Self-Driving Entity laid out in the UK Automated Vehicles Act to designate responsibility for EAI operation. Policymakers must also manage the tension between fully autonomous physical systems and GDPR Article 22's recourse to human intervention. For example, policymakers could prohibit EAI deployment in situations where such recourse to human decision-making would be impossible or outline scenarios in which humans do not have this Article 22 right. At the same time, policymakers must work with technologists to determine when a manufacturer should be held accountable for errant EAI actions (e.g. when training is deemed insufficient for deployment in specific environments, or when new models are released but manufacturers do not make safety-relevant over-the-air updates available).

## 4.5  Plan and prepare for the transformative economic and social effects of EAI

Policymakers at the national and international levels should draft legislation to prepare safety-net or assistance programs for people whose labor is replaced by EAI systems. Basic proposals have been floated concerning UBI [170], or even universal basic compute (UBC), whereby people are guaranteed access to and use of AI or EAI systems [171]. However, these proposals remain very sparse and abstract. Policymakers should create draft frameworks and attempt to form early consensus now, as highly advanced EAI models and widespread labor displacement may arrive in the near future. Policymakers should specifically address who will be eligible to claim these social assistance packages and under what conditions (e.g. what type of proof will be required to demonstrate that an individual lost their job as a direct result of EAI automation). Reskilling programs are another potential policy avenue; however, these worker retraining programs may face limitations in the face of AI and EAI that automate an increasing number of jobs [172].

Similarly, policymakers must better prepare for transformative social effects [96]. Given the capital-intensive nature of EAI systems, it is plausible that EAI power and access could be concentrated in the hands of a select few [98]. Policymakers should draft options to combat this social power concentration, perhaps through targeted taxation mechanisms [173–175]. Policymakers should also fund research on mitigating adverse emotional dependencies between EAI systems and humans. EAI deployment is ultimately (for now) a human decision, so policymakers should consider whether some domains should be entirely off-limits for EAI interaction. Organizations such as the OECD, GPAI, or the nascent UN AI Panel and Dialogue should prioritize action on these pressing social issues, as EAI will impact people worldwide, not just in today's robotics hotspots.

# 5 Discussion and Limitations

Our analysis has several limitations, and key counterarguments to our main claims warrant further attention. We hope this discussion helps pave the way for future work on this exciting topic.

**Geographical limitations.** We focus on policies from the US, UK, and EU given the authors' location and expertise. However, future analyses should extend to other geographies, particularly China, India, Japan, and Korea. Conversations about EAI safety should involve robust global representation, as notions of safety shift across geographies and cultures [176].

**Application limitations.** This paper primarily focuses on civilian applications of EAI; however, military and law enforcement EAI applications also demand urgent policy action. The development and deployment of weaponized EAI systems are expected to continue growing over the coming months and years [177]. This could markedly lower barriers for non-state actors to cause significant damage and for political leaders to suppress dissent with fleets of embodied agents [78].

**Methodological challenges.** EAI is a vast and rapidly growing field, and we omit many exciting areas of discussion due to space limitations. For example, the potential for high-fidelity virtual simulations and video-based learning to revolutionize EAI training merits its own review [70]. We also recognize that the EAI field is inspired by decades of engineering, computer science, and human-robot interaction research. We had to make difficult decisions in creating our main categories of risk, which appeared in conversations with experts and in our literature review. We hope these categories serve as a starting point for more robust discussions about key areas of EAI risk.

**Market forces and societal pressures.** We recognize that good policy does not always mean more regulation; active intervention may not be necessary if market or social pressures naturally lead to safer EAI. For example, manufacturers or countries with strong safety protocols might be seen as having a commercial advantage [178]. However, as with virtual AI systems, market forces may lead to race dynamics, less government oversight, and exacerbated risks [179].

**Technical solutions and EAI harm.** Technical solutions (e.g., alignment, unlearning, etc.) will play a crucial role in mitigating EAI risks. However, technical solutions may only be established and implemented effectively after EAI risks emerge at scale, as technical solutions are often responsive to demonstrated needs [180]. Technical solutions will also likely not address all the risks mentioned above—especially those related to economic and social risks. A more balanced and pragmatic approach is required, one that combines the best aspects of technical and non-technical solutions [49].

**Overlap of EAI risks with LLM risks.** Skeptics could argue that if we make LLMs safe, we will solve EAI risks. We acknowledge there is considerable overlap between the relevant risks [181], but many EAI risks are modified or magnified at the application and organizational level [50]. EAI creates immediate and pervasive risks by virtue of existing in the physical world that could grow with scale. For example, an EAI could impair a baby's healthy development [182] or be hacked and cause a deadly crash in ways inapplicable to purely virtual AIs [183]. Improving LLM safety alone is helpful but would fail to cover these—and many other—critical EAI scenarios.

# 6 Conclusion

The EAI field is rapidly advancing, driven by increasing hardware investment, breakthroughs in LLMs and LMMs, and quickening deployment. These trends will likely accelerate in the coming years. However, policymakers around the world have thus far neglected EAI governance even though associated risks have gradually transitioned from the realm of science fiction to the real world.

We have argued that policymakers should encourage the development of effective benchmarks, evaluations, and safety protocols for the responsible deployment of EAI; ensure safety certification for a range of EAI form factors, capabilities, and operational domains; reevaluate liability paradigms; confront labor displacement; and address larger societal issues, such as human-EAI attachment. These recommendations provide first steps that can guide and encourage safe EAI innovation with minimal downside. Many other risks necessitate minor tweaks or adjustments to existing policies—for example, preventing privacy violations from EAI systems in public will require integration with existing laws such as the MR and GDPR. Zooming out, research funding for EAI safety should be expanded, and policymakers should collaborate with AI researchers and EAI developers to translate research findings into policy advice.

## Acknowledgements

Jared Perlo is grateful to the Centre for the Governance of AI, where he was a Winter Fellow, and the Centre pour la Sécurité de l'IA for making this research project possible. We would like to thank Aaron Prather, Aidan Homewood, Amelia Michael, Connor Aidan Stewart Hunter, Edward Kembery, Jenny Read, Keegan McBride, Krzysztof Bar, Kyler Zhou, Liam Patell, Markus Anderljung, Marta Ziosi, Nora Amman, Pierre Sermanet, Roeland P.-J. E. Decorte, Shaoshan Liu, Simon Mylius, Todor Davchev, Umair Siddique, Yohan Mathew, and Yunzhu Li for their valuable input and collaboration.

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
