# OpenReview forum: "Emerging Risks from Embodied AI Require Urgent Policy Action"
_NeurIPS.cc/2025/Position_Paper_Track — NeurIPS 2025 Position Paper Track_

### Official Review · Reviewer_a1j5 · 2025-07-31

**Significance:** 4
**Presentation:** 3
**Rating:** 7
**Confidence:** 4

**Summary:**

This paper argues that fast‑moving Embodied AI (EAI) brings physical, informational, economic and social risks that current rules cannot handle, so researchers and policymakers need to fix the gaps now. It offers a four‑part risk taxonomy, maps how US‑UK‑EU law covers each risk, and shows huge holes on labour displacement and social impact. The authors propose five policy tracks: invest in EAI safety research, mandate verification and certification, expand industry standards, clarify liability, and prepare social safety nets. They rebut market‑only and “LLM safety equals EAI safety” views and call for joint action before a crisis forces rushed laws.

**Strengths:**

1. Clear structure linking risk taxonomy, policy heat map and concrete recommendations
2. Uses up‑to‑date cases (AV STEP, UK AV Act) that make the policy review timely
3. Anticipates counter‑arguments and answers them, which strengthens persuasiveness

**Weaknesses:**

1. Economic and social proposals stay high level, lacking costed scenarios or metrics
2. Benchmarks focuses on autonomous vehicles analogies, lack of analysis on other popular robots

**Questions:**

1. Do you have a quantitative model to show how different certification regimes cut accident or job‑loss risk, and could you share assumptions?
2. How would mandatory “black box” logging respect privacy in homes and hospitals; would tiered access or on‑device redaction work?

**Alternative Position:**

Yes, and alternative positions are well-considered and addressed by the argument

**Author Identification:**

No.

**Context:**

3

**Discussion:**

4

**Ethics:**

["NO or VERY MINOR ethics concerns only"]

**Position:**

Yes, the paper argues for or against a position related to machine learning.

**Support:**

3

**Thoroughness:**

4

---

### Official Review · Reviewer_ikcM · 2025-08-12

**Significance:** 4
**Presentation:** 3
**Rating:** 7
**Confidence:** 4

**Summary:**

1. The paper proposes a four-part risk taxonomy for EAI, surveys existing policies, offers policy pathways, and suggests investments in safety research, verification, standard updates, liability clarification, and economic/social impacts planning.
2. It surveys existing policies/standards across the US/UK/EU and classical robotics/AVs/virtual agents, including a “heat-map” of coverage gaps.
3. Mandates verification/validation/certification with model cards and audits; update and extend standards.
4. Finally clarifies liability for fully autonomous systems; and plans for economic/social impacts.

**Strengths:**

1. Actionable agenda:
    Certification with model cards/audits; explicit call for black boxes and jailbreak/cybersecurity protocols.
2. Useful Synthesis:
    Crisp four-way risk taxonomy and a policy coverage heat-map that highlights where AV/robotics rules help—and where they don’t.
3. Specific Governance levers: ASDE/NUiC ideas as transferable precedents; liability clarifications for fully autonomous systems.
4. Benchmarks focus:  The evaluations done are beyond virtual AI test suites.

**Weaknesses:**

1. Operationalization gap: proposes certification/benchmarks but lacks draft metric suites or test protocols.
2. Regulatory mapping: identifies “who regulates?” as a gap but stops short of concrete institutional designs (e.g., which US agency, scope, and authorities).
Evidence depth: several motivating incidents are secondary-source reports; more systematic incident data or case studies would strengthen claims.
Economic claims: Displacement (or) inequality arguments are plausible but would benefit from quantitative scenarios or policy costing.
Geographic scope: limited to US/UK/EU; omits China/Japan/India despite acknowledging this in limitations (Future Work?)

**Questions:**

1. Which minimum test battery do you propose and how would pass/fail thresholds be set per operational domain?
2. What safeguards/retention policies avoid surveillance overreach while enabling incident forensics for home/clinic robots?
3. Can you sketch 3–5 near-term EAI benchmarks (tasks, metrics, eval harnesses) that correlate with real-world risk reduction, not just task success?
4. How should fault apportion when software and hardware come from different vendors and models are updated over-the-air?

**Alternative Position:**

Yes, and alternative positions are well-considered and addressed by the argument

**Author Identification:**

No.

**Context:**

3

**Details Of Ethics Concerns:**

Maybe consider a short note clarifying any use of third-party incident reports or case studies referenced in figures/tables

**Discussion:**

4

**Ethics:**

["NO or VERY MINOR ethics concerns only"]

**Position:**

Yes, the paper argues for or against a position related to machine learning.

**Support:**

3

**Thoroughness:**

4

---

### Official Review · Reviewer_Vb6u · 2025-08-12

**Significance:** 3
**Presentation:** 4
**Rating:** 7
**Confidence:** 4

**Summary:**

This paper is a call to action on the urgent need for stronger policies around embodied AI (EAI): AI systems with a physical presence like robots, autonomous vehicles, and other machines. The paper presents a clear taxonomy of EAI risks: physical, informational, economic, and social, then creates a heat map of those risks against existing laws and standards to show the gaps. The authors support their arguments with real-world examples and policy comparisons, arguing that current frameworks are too limited. They present practical recommendations (like mandatory safety certifications of EAI systems), and a strong argument for involving ML researchers in governance.

The position of the paper is that: EAI is advancing fast, the risks are real, and both policymakers and the research community need to act before those risks scale.

**Strengths:**

-  The paper presents a good taxonomy of physical, informational, economic, and social risks of EAI.
- The paper presents a policy heat-map of coverage gaps, clearly discussing the large gaps.
- The authors clearly consider and discuss the alternative views.
- The paper offers actionable policy recommendations that can be directly connected to current technical and governance debates.

**Weaknesses:**

- The paper could provide concrete details on how proposed benchmarks, certification systems, and liability regimes would be implemented and enforced across different industries.
- The authors should distinguish between near-term risks (based on current EAI capabilities) and long-term risks, to prioritize policy recommendations and actions.
- The paper doesn't expand beyond the US, UK, and EU and should include perspectives from countries with significant robotics and AI activity, such as China and Japan.

**Questions:**

- The paper focuses on policy gaps but doesn't give enough attention to real-world challenges like political resistance. How do you see this being addressed and what is the role of AI researchers?
- You assume policymakers and ML researchers will coordinate, do we have evidence or precedents to support this, and what could AI researchers do if this coordination fails?

**Alternative Position:**

Yes, and alternative positions are well-considered and addressed by the argument

**Author Identification:**

No.

**Context:**

3

**Discussion:**

3

**Ethics:**

["NO or VERY MINOR ethics concerns only"]

**Position:**

Yes, the paper argues for or against a position related to machine learning.

**Support:**

4

**Thoroughness:**

3

---

### Note · Authors · 2025-09-05

**1-10 Additional Comments:**

We really appreciated the opportunity to submit to this track. We think having a position paper track is a great addition for NeurIPS, and it seems slated to spur and spark important discussions within the community. In our case, the existence of this track led us to form an author team and collaboration that otherwise would not have happened, as we were all working on projects in the same area but had not previously been in touch.

**1-11 Submit Again:**

Definitely yes

**1-1 Submission Process:**

5

**1-2 Next Year:**

We thought this year's process was rather smooth, all things considered. On the administrative side, we would like to see more information up front about how the reviews and surveys will be conducted. It would be helpful to understand how reviewers and authors will communicate and iterate about the suitability of a paper for this track. On the content side, it could be helpful to see a list of topics that were represented in this year's position paper track to gauge general paper and topic relevance for future submissions.

**1-3 Future Development:**

To develop and improve the track, it would be helpful to know the broad categories that position papers fell into this year. Perhaps there could be incentives or prizes for most heterodox-but-interesting/useful paper. Perhaps there could be a concerted effort to match technical and position papers from similar topic areas to foster cross-author collaboration and exchange.

**1-4 Interest:**

["Panel discussions with other position paper authors", "Structured debates on controversial topics", "Workshops for developing position papers", "Mentorship programs for early-career researchers"]

**1-4 Other Interest:**

As mentioned above, it could be interesting to have panel discussions or workshops bringing together technical and position paper authors who address or cover similar material. This could be a great way of sparking more discussion and collaboration between researchers who otherwise wouldn't interact (i.e. bridging two relatively siloed groups).

**1-5 Thoughtful:**

9

**1-6 Supportive:**

7

**1-7 Technical Aspects Versus Position:**

6

**1-8 Gate Keeping:**

9

**1-9 Camera Ready Changes:**

We would like to make several small tweaks that should improve the quality of the paper. To be very specific to our paper, we want to include a refined version of our analysis of existing policies for embodied AI, for example by adding in a small section on drone regulation in the EU, UK, and US. We have also added several high-quality sources to back up claims in our taxonomy, so we would like to add in those citations to our body and bibliography. Per review feedback, we would also want to include more specifics about potential benchmarking and evaluation schemes for different modalities and forms of EAI, as it was noted in several reviews that this area would benefit from more detail. Likewise, we would include a piece about the importance of political economy in being able to usher in policy changes, as political winds and climate are likely a strong reason why action has not been taken so far. We greatly appreciated the reviewers' feedback and think these shifts will substantially improve our paper.

---

### Meta-Review · Area_Chair_ioyk · 2025-09-01

**Rating:** 7
**Confidence:** 4

**Strengths:**

+ The paper has a clear position and proposes actionable suggestions for mitigating the risks of embodied AI.

+ The paper also clearly discusses the alternative view.

**Weaknesses:**

- Some details can be provided on the implementation suggestions of the proposed solutions (benchmarks, liability regimes, etc).

**Questions:**

It is an interesting topic to discuss whether the policy and action recommendations would be different for short-term and long-term risks.

**Ethics:**

No issue found.

**Thoroughness:**

4

---

### Decision · Program_Chairs · 2025-09-26

Accept